# Dietary Intake and Physical Activity of Thai Children and Adolescents with Type 1 Diabetes Mellitus

**DOI:** 10.3390/nu14235169

**Published:** 2022-12-05

**Authors:** Sriwan Thongpaeng, Preeyarat Sorncharoen, Lukana Preechasuk, Jeerunda Santiprabhob

**Affiliations:** 1Siriraj Diabetes Center of Excellence, Faculty of Medicine Siriraj Hospital, Mahidol University, Bangkok 10700, Thailand; 2Division of Endocrinology and Metabolism, Department of Pediatrics, Faculty of Medicine Siriraj Hospital, Mahidol University, Bangkok 10700, Thailand

**Keywords:** diabetes mellitus, type 1, eating, exercise, children and adolescents, calcium, dietary fiber, Thailand

## Abstract

Appropriate dietary intake and physical activity (PA) are essential for glycemic control and optimal growth in youth with type 1 diabetes (T1D). Thus, this study aimed to compare dietary intake and PA between youth with T1D and healthy controls. One hundred Thai youth with T1D and 100 age-matched healthy participants were recruited. A 3-day food record was completed and converted into nutrient intake data. PA data were collected via interview. Participants with T1D had a significantly higher mean ± SD carbohydrate (50.8 ± 6.8% vs. 46.2 ± 7.5%, *p* < 0.01), lower fat (32.4 ± 5.9% vs. 35.9 ± 6.4%, *p* < 0.01), and lower protein (16.8 ± 2.6% vs. 17.9 ± 3.5%, *p* = 0.01) intake compared to controls. Fifty percent of T1D participants and 41% of control participants consumed saturated fat more than recommendations (*p* = 0.20). Participants with T1D had a higher median (IQR) calcium intake compared to controls (474 (297–700) vs. 328 (167–447) mg/day, *p* < 0.01). Both groups consumed less fiber and more sodium compared to recommendations. Both groups had inadequate PA. Participants with T1D had significantly less PA compared to controls (25 (13–48) vs. 34 (14–77) minutes/day, *p* = 0.04). In addition to the need for counseling that promotes consumption of more dietary fiber and calcium and less saturated fat and sodium, the benefits of performing regular exercise need to be emphasized among youth with T1D.

## 1. Introduction

Type 1 diabetes (T1D) is one of the most common chronic diseases among children. The incidence of T1D has been increasing globally [1,2]. In Thailand, the incidence of T1D during 1991–1995 was 1.65 per 100,000/year [3], and the prevalence of T1D diagnosed before the age of 18 years among all patients with diabetes in the Thailand Diabetes Registry Project in 2003 was 2.07% [4]. The cornerstone of T1D management includes insulin treatment, self-monitoring of blood glucose, and nutritional management [5,6,7]. Adequate physical activity is also recommended to improve physical fitness and cardiovascular function [7,8].

Dietary intake recommendations for youth with T1D are based on healthy eating principals suitable for all children with the aim to optimize growth and development, improve glycemic control, and prevent acute and chronic complications [6]. The percentage of macronutrient distribution should depend on individualized health conditions. The International Society for Pediatric and Adolescent Diabetes (ISPAD) guideline recommends macronutrient distribution as follows: carbohydrate 45–50% of energy, fat < 35% of energy (saturated fat < 10%), and protein 15–20% of energy (Table 1) [6].

A large multicenter cohort study of youth with diabetes in the United States showed that the majority of youth with diabetes did not meet the dietary recommendations [9]. An audit of dietary intake among Australian children with T1D found lower carbohydrate and higher total sugar, fat, saturated fat, and protein intake compared to the recommendation [10]. A literature review of dietary adherence among youth with T1D revealed a higher than recommended intake of fat and saturated fat, and a lower than recommended intake of fruits, vegetables, and whole grains in many studies [11]. 

ISPAD guideline also recommends 60 min or more of physical activity each day for children and adolescents with T1D aged from 6 to 18 years. The physical activity should include (1) moderate to vigorous intensity aerobic activity, (2) muscle strengthening activity, and (3) bone strengthening activity [8]. These recommendations are the same as those recommended for healthy children and adolescents [8]. However, previous studies in youth with T1D reported lower physical activity compared to both the recommendation and healthy controls [12,13,14].

Data specific to dietary intake and physical activity in children and adolescents with T1D in Thailand are scarce. Accordingly, the aim of this study was to compare dietary intake and physical activity between children and adolescents with T1D and healthy age-matched controls. In addition to improving our understanding of these two important T1D management-related parameters, the findings of this study will be useful for developing strategies to improve treatment outcomes and quality of life among Thai youth with T1D. 

## 2. Materials and Methods

This study is part of a cross-sectional study investigating bone mineral density among children and adolescents with T1D [15] that was conducted at the Division of Endocrinology and Metabolism, Department of Pediatrics, Faculty of Medicine Siriraj Hospital, Mahidol University, Bangkok, Thailand, during March 2016 to January 2018. The protocol for this study was approved by the Siriraj Institutional Review Board (SIRB) (COA no. Si 145/2016). Child assent to participate and parental written informed consent were obtained in children aged 7 years to <18 years, and written informed consent was obtained from adolescents aged 18 years or older prior to entering the study.

### 2.1. Participants 

One hundred children and adolescents with T1D who routinely attended the Diabetes Clinic of the Department of Pediatrics, Siriraj Hospital, were recruited. The inclusion criteria were as follows: (1) diagnosed with T1D for at least 1 year; (2) aged 5–20 years; (3) had not taken drugs that affect bone mineral density (BMD) (e.g., calcium, vitamin D, multivitamin, or corticosteroid) within 6 months prior to enrollment; (4) absence of any diseases that affect vitamin D levels or bone density (e.g., uncontrolled hyperthyroidism, malabsorption syndrome, celiac disease, chronic liver disease, impaired renal function, and bone diseases) [15]. 

In our department, children and their caregivers receive inpatient diabetes self-management education (DSME) for 5–7 days at diagnosis or first referral visit. Nutritional education is taught by nutritionists. The topics include food groups, food exchange, carbohydrate counting, healthy diet, matching insulin dose to carbohydrate intake, and individualized meal planning at home and at school. Pediatric endocrinologists and nutritionists work together to define energy intake and macronutrient distribution. As a general guide, macronutrient distribution in general patients is recommended to be carbohydrate 50% of energy, fat 30% of energy, and protein 20% of energy. 

The control group comprised 100 healthy children and adolescents aged 5–20 years who had no diseases and who took no drugs that interfere with vitamin D levels or bone density. Healthy participants were matched with T1D patients by gender, age (±2 years), BMI Z-score (±0.5), and pubertal status. Control group children and adolescents were recruited from schools located near Siriraj Hospital [15].

### 2.2. Physical Data Collection and Biochemical Measurements

Weight, height, body mass index (BMI), breast Tanner stage, genital Tanner stage, testicle size, and pubic hair Tanner stage were assessed [15]. Height and weight Z-scores were calculated using the INMU-NutriStat software program (Mahidol University, 2002). Due to the lack of a national BMI reference, BMI Z-scores were analyzed using World Health Organization (WHO) 2007 growth reference data [16]. HbA1c was determined by turbidimetric inhibition immunoassay (Integra 400 analyzer; Roche Diagnostics). HbA1c were obtained on the day of enrollment. 

### 2.3. Body Composition Analysis 

Body composition including percentage of total fat and total amount of lean mass, was measured using a dual energy X-ray absorptiometry (DEXA) scan (Lunar Prodigy; GE Healthcare, Chicago, IL, USA). 

### 2.4. Diabetes Treatment, Health Insurance Scheme, and Socioeconomic Status of Participants 

Data on insulin treatment regimen, frequency of self-monitoring of blood glucose (SMBG), and health insurance schemes among participants with T1D were collected. The three main health insurance schemes in Thailand include (1) the Civil Servant Medical Benefit (CSMB) scheme for government employees and their dependents, (2) the Social Health Insurance (SHI) scheme for private sector employees, and (3) the Universal Health Coverage (UC) scheme for the remaining Thai population [17]. All health insurance plans cover the cost of insulin for patients with diabetes. However, the cost of glucose test strips was not covered by any of the health insurance schemes. It was only after October 2018 that patients with T1D using the UC scheme who participated in the Thai Type 1 Diabetes and Diabetes Diagnosed Before Age 30 Years Registry, Care and Network (T1DDAR CN) could acquire reimbursement for glucose test strips up to 4 strips/day [18]. Educational level of parents and monthly household income were recorded. 

### 2.5. Dietary Intake Assessment

On the date of study enrollment, participants were taught to estimate portion size using a picture guide of Thai food exchange lists, and using household measures such as tablespoon, teaspoon, ladle, glass, or cup. Participants were asked to record dietary intake for 3 days including two weekdays and one weekend day. Once the dietary intake record was completed, participants were instructed to return it by mail to the study team. If the data from the food record was incomplete or participants failed to mail it back, dietitians obtained dietary recall by telephone. The average dietary intake from the 3-day record was converted into nutrients using the INMUCAL–Nutrients version 3 software program. This program was developed by the Institute of Nutrition, Mahidol University, Nakhon Pathom, Thailand. Carbohydrate, protein, total fat, saturated fat, and total sugar, including natural and added sugar, were expressed as percentages of total energy (%TE). The nutritional intake of patients was compared with the nutritional recommendation from ISPAD (shown in Table 1).

### 2.6. Physical Activity Assessment 

On the day of study enrollment, participants were interviewed to obtain information about their physical activity during a typical week within the previous month using the Global Physical Activity Questionnaire (GPAQ, version 2, World Health Organization) [19], Thai version (Thai National Health Examination Survey IV, 2008–2009). The questionnaire consists of 3 domains, including activity at work, travel to and from places, and exercise. The intensity, duration per session, and frequency of activities in a typical week were recorded. Only activities with a duration of at least 10 min were recorded. The duration of activities per week was divided by 7 for conversion to duration per day. The activities from GPAQ can be categorized as vigorous work, moderate work, travel, vigorous exercise, or moderate exercise. The intensity of physical activity was reported as a metabolic equivalent of task (MET) score. One MET is defined as the energy cost of sitting quietly, and is equivalent to a caloric consumption of 1 kcal/kg/h. 

### 2.7. Statistical Analysis

The data were analyzed using SPSS Statistics for Windows, version 20.0 (SPSS, Inc., Chicago, IL, USA). Continuous data are reported as mean ± standard deviation (SD) for normally distributed data, and as median and interquartile range [IQR] for non-normally distributed data. Categorical data are given as number and percentage. Comparisons between groups were conducted using independent samples t-test (normally distributed data) and Mann–Whitney U test (non-normally distributed data) for continuous data, and using Fisher’s exact test or chi-square test for categorical data. A *p*-value < 0.05 was considered to be statistically significant. 

## 3. Results

### 3.1. Clinical Characteristics, Body Composition, and Biochemical Data 

Demographic, anthropometric, clinical, and biochemical characteristics of the entire study population, including the comparison between male and female study participants, are shown in Table 2. There were no significant differences in parents’ educational level, household income, age, gender, BMI, or pubertal stage between youth with T1D and controls. The health insurance scheme in the T1D group was UC (67%), CSMB (24%), and self-payment (9%). Among the T1D group, the median [IQR] duration of diabetes was 5.8 years [2.97–9.07]. Sixty-one percent of participants with T1D used a basal bolus insulin regimen, 38% used a modified conventional insulin regimen (3 injections/day), and 1% used premixed insulin twice daily. For prandial insulin, all patients used a rapid-acting insulin analogue. For basal insulin, intermediate-acting insulin was used in the modified conventional insulin regimen, and a long-acting insulin analogue was used in the basal bolus insulin regimen. The mean insulin usage was 1.16 ± 0.30 units/kg/day. All patients used glucometers to monitor their blood glucose. The frequency of self-monitoring of blood glucose per day was ≤1 (10%), 2 (14%), 3 (41%), and ≥4 (35%). There was no continuous glucose monitoring (CGM) or continuous subcutaneous insulin infusion (CSII) usage in this study. Lean body mass and % total fat was not significantly different between youth with T1D and the controls, and also not significantly different by gender between groups. Interestingly, height Z-score was significantly lower in the T1D group than in the control group ((*p* = 0.003). However, the difference was only statistically significant among females (*p* = 0.01), not in males (*p* = 0.132).

### 3.2. Dietary Intake

Dietary intake was assessed by food record (66% in T1D group, 58% in control group) and dietary recall by telephone (34% in T1D, 42% in controls). Youth with T1DM had significantly higher energy intake than youth without T1D. The T1D group had a higher percentage of energy from carbohydrate, a lower percentage of energy from fat and protein, and higher calcium intake compared to the control group. However, the energy from saturated fat was not significantly different between groups. Carbohydrate and fat distribution were significantly different between youth with T1D and controls among males, but not among females (Table 3). The mean of macronutrient distribution in both the T1D and the control group was close to the ISPAD recommendation range.

The proportion of patients who had fat distribution greater than 35% of total energy was significantly lower in participants with T1D than in controls (*p* < 0.01), and this difference between groups was even more pronounced among males (*p* < 0.01) (Table 4). In both youth with T1D and controls, sodium intake was higher than the World Health Organization recommendation (sodium < 2000 mg/day) [20], and fiber intake was lower than the ISPAD recommendation [6]. Using Thai dietary intake recommendations (calcium intake >800 mg/day) [21], both groups had a high proportion of unmet calcium requirement; however, the control group had a significantly higher proportion of inadequate calcium intake compared to the T1D group (*p* = 0.02) (Table 4). 

Concerning eating patterns, 11% of the healthy group reported skipping their breakfast, 4% skipped lunch, and 3% skipped dinner (17% in total). In contrast, no participants with T1D reported skipping any meals. 

### 3.3. Physical Activity

Physical activity compared between participants with T1D and the control group during a typical week within the previous month is shown in Table 5. The majority of participants in both groups did not perform vigorous work or travel. Youth with T1D had significantly lower median [IQR] travel activity (*p* = 0.01) and overall physical activity compared to controls (*p* = 0.04). Regarding exercise, a significantly lower proportion of T1D participants performed moderate to vigorous exercise ≥150 min/week (*p* = 0.02). In the male group, participants with T1D spent significantly less time on travel activity (*p* = 0.04), overall physical activity (*p* = 0.01), and overall exercise (*p* < 0.01), and had significantly lower total MET value (*p* < 0.01) compared to participants in the control group. In the female group, there was no significant difference in physical activities between participants with T1D and controls, and both female groups had a low level of physical activity. 

## 4. Discussion

Our study demonstrated differences and similarities in certain aspects of dietary intake and physical activity between Thai youth with and without T1D. Youth with T1D had relatively healthier dietary intake than youth without T1D in terms of higher calcium intake and regular meal consumption. However, a high proportion of both groups consumed less calcium and dietary fiber and more sodium than the recommended intake. In addition, a comparable proportion of both groups consumed saturated fat >10% of total energy. Male youth with T1D had lower physical activity and exercise compared to healthy male participants.

Concerning energy intake, we found that youth with T1D had higher energy intake compared to healthy participants. Youth with T1D in our study had macronutrient distribution within the ISPAD recommendation range [6]. They had higher energy intake from carbohydrates, and lower energy intake from protein and fat compared to healthy participants. These differences in carbohydrate and fat intake were even more pronounced in the male group. Both participants with T1D and the controls had saturated fat consumption at the upper limit of recommendation. Our study demonstrated different results compared to previous studies. A study of adolescent girls with T1D from the United Kingdom [22] and a study of adolescents from the United States [23] found no differences in energy intake between adolescents with and without T1D. However, data from Australia revealed that children with T1D aged 4–13 years had higher energy intake compared to data from a national survey, but there was no difference in energy intake between adolescents with and without T1D aged 14 years or older [10]. Regarding macronutrient distribution, compared with studies from the United States and Europe [9,10,23,24], western youth with T1D consumed slightly lower energy from carbohydrate and higher energy from fat than the youth in our study. Furthermore, the percentage of saturated fat and total sugar consumption in our study was lower than in Western studies [9,10].

The differences in dietary intake between youth with and without T1D found in our study might be explained by the nutritional education that youth with T1D and their care givers received at first diagnosis, and the fact that their dietary habits are regularly reviewed and discussed at each follow-up visit. Age-specific energy intake and portion of carbohydrate were calculated by endocrinologists and nutritionists for every patient. Balanced nutritional diet and quality of food choice were both emphasized. The higher fat intake in the control group might be influenced by a nutritional transition that is occurring in Thailand. Thai traditional dietary intake that relies heavily on rice, aquatic animals, vegetables, and herbs is being replaced by a diet containing a higher proportion of fat and animal meat, and a lower proportion of vegetables and fruits. Kosulwat, et al. reported an increasing trend of fat consumption in Thailand during 1960–1995 [25]. Despite having higher energy consumption, the T1D group had similar BMI and body composition as the control group. This might be partly explained by the fact that youth with T1D in our study had fair glycemic control in which energy loss in the urine due to a high glycemic level could occur. In addition, underestimating energy intake in the control group due to an unfamiliarity with food portions could also influence the difference in the energy intake between participants with T1D and the control groups.

Youth with T1D in our study had significantly higher calcium intake compared to control group youth. However, this finding was in contrast to some studies included in a previous systematic review [26]; Gil-Díaz, et al. found no difference in calcium intake between participants with T1D and the control group among the six studies included in their systematic review. Higher calcium intake in the T1D group in our study might be explained by higher consumption of milk because we generally recommend 2–3 portions of milk per day for children and adolescents with T1D. However, only 17% of youth with T1D and 6% of youth without T1D consumed enough calcium to satisfy the Thai calcium intake recommendation [21]. This problem has been reported in many studies [26]. Since low bone mass and increased fracture rate have been recognized as complications of T1D [27], adequate consumption of calcium, the critical determinant of bone mineralization, should be ensured. Calcium-rich foods in Thailand, such as milk, soybean milk, dairy products, leafy green vegetables, small fish with edible bone, and white hard tofu, should be promoted to both T1D and non-T1D youth.

In our study, participants with and without T1D consumed very low amounts of fiber and high amounts of sodium. Only 3% of the T1D group and 2% of the control group met the recommendation for fiber consumption (14 g fiber/1000 kcal), and 81% of T1D participants and 75% of the controls consumed sodium higher than 2000 mg per day. Dietary fiber promotes digestive health via its modulation of laxation and fermentation, and its effect on gut microbiota [28]. Concerning glycemic control, Nansel, et al. found dietary fiber to be associated with better glycemic control in youth with T1D as indicated by higher 1,5-anhydroglucitol and lower mean glucose from continuous glucose monitoring, as well as lower glucose variability as indicated by lower standard deviation and mean amplitude of glycemic excursions [29]. These glycemic benefits of fiber might be caused by a reduction in the rate of glucose absorption via its beneficial viscosity properties [30,31]. 

Regarding sodium intake in patients with T1D, Anderson, et al. reported association between higher dietary sodium intake and vascular smooth muscle dysfunction [32]. The Finnish Diabetic Nephropathy Study showed nonlinear association between urinary sodium excretion and all-cause mortality [33]. Even though the benefits of dietary fiber and the detrimental effect of high sodium intake are well recognized, inadequate fiber consumption and high dietary sodium intake have been widely reported in youth with and without T1D, as well as in the present study [9,11,23,24,34,35,36]. Puwastein et al. found a low concentration of dietary fiber in rice, which is the staple Thai food, and a wide variation of fiber content in vegetables (ranging from 1 g to 13.6 g in 100 g of vegetables) and fruits (ranging from 0.3 g to 8 g in 100 g of fruits) in Thailand [37]. Anderson, et al. found association between higher sodium intake and higher intake of takeaway food and processed snack foods [32]. Therefore, the strategies that we propose to overcome these dietary challenges include replacing white rice with brown rice, promoting consumption of vegetables and fruits with high fiber content, and decreased consumption of takeaway food and processed snack foods to both T1D and non-T1D children and adolescents.

We observed regular meal consumption in youth with T1D, while 17% of controls skipped meal consumption, especially breakfast. This finding suggests that participants with T1D acknowledge the importance of eating a meal after insulin injection. A regular meal pattern was associated with better glycemic control in youth with T1D in previous studies [24,38]. 

Physical activity was reported to benefit HbA1c, BMI, lipid profile, blood pressure, and psychological well-being in children and adolescents with T1D [39,40,41]. However, inadequate physical activity was reported in both children and adolescents with T1D and their healthy peers [12,13,14,42]. Our study found a significantly lower level of travel activity, vigorous exercise, and overall physical activity in males with T1D compared to males without T1D and a very low level of physical activity in females with and without T1D. Only 50% of males with T1D and 18% of females with T1D performed moderate to vigorous exercise ≥150 min/week. The most common barrier to physical activity in children and adolescents with T1D was fear of hypoglycemia, followed by loss of control over diabetes, school schedule, weather, and low fitness level [13,43,44]. Physical activity can cause glucose fluctuation, either hypoglycemia or hyperglycemia, depending on the type, duration, and intensity of physical activity. Management to optimize blood glucose level includes adjusting prandial and/or basal insulin, nutritional planning, and frequent glucose monitoring [8,45]. Therefore, to overcome the common barriers to physical activity, an individualized blood glucose management plan for physical activity should be developed for each patient. Furthermore, patients and their parents should keep detailed records of their physical activity, insulin use, food intake, and glucose response to improve the development of a much more individualized plan [8]. Not only youth with T1D, but also healthy adolescents have inadequate physical activity. A pooled analysis of population-based surveys found 81% of adolescents globally had insufficient physical activity (77.6% of males, 84.7% of females). Females were less active than boys across all regions, including Thailand [42]. Thus, in addition to the solutions proposed to remedy the challenges associated with youth with T1D, additional strategies should be developed to break down the barriers reported by healthy adolescents that include life transition periods, attitude and motivation relative to physical activity, friend and family influence, and environmental opportunities [46]. 

This is the first study to investigate the dietary intake and physical activity of Thai youth with T1D compared with their healthy age-matched peers in Thailand. However, this study has some limitations that need to be discussed. First, some study participants did not return their food intake record, thus, the dietary intake information was taken from participant recall via telephone interview. Second, we recruited healthy participants from schools located in Bangkok, the capital city of Thailand. It is, therefore, possible that the characteristics of these children may not reflect the characteristics of the children living in rural areas of Thailand. Third and last, we were not able to clarify and report the percentages of monounsaturated fat, polyunsaturated fat, and added sugar due to limitations of our nutrient conversion program. 

## 5. Conclusions

Although the distribution of dietary macronutrient intake in both the T1D and control groups complied with the recommended intake, the majority of both T1D and control participants had low fiber, low calcium, and high sodium intake compared to the recommendation. Moreover, male youth with T1D had lower physical activity and exercise compared to healthy participants. Of concern, both T1D and control females were found to be lacking an active lifestyle. Therefore, healthcare professionals should counsel children and adolescents to eat a healthy balanced diet, consume more dietary fiber and calcium, and eat less saturated fat and sodium. Furthermore, the benefits of exercise should be addressed, and being physically active should be encouraged both among children with T1D and those without. Exercise-related diabetes self-management education needs to be provided to all children and adolescents with T1D, in order for them to exercise safely and without fear of hypoglycemia. 

## Figures and Tables

**Table 1 nutrients-14-05169-t001:** Nutritional recommendations for children and adolescents with type 1 diabetes mellitus from the International Society of Pediatric and Adolescent Diabetes (ISPAD).

	Recommendation for DM
Carbohydrate	45–50%TE
Protein	15–20%TE
Fat	30–35%TE
Saturated fat	<10%TE
Sugar	<10%TE
Dietary fiber	14 g/1000 kcal/day
Sodium	As recommendation for general population

%TE: percentage of total energy.

**Table 2 nutrients-14-05169-t002:** Demographic, anthropometric, clinical, and biochemical characteristics of the entire study population, and compared between male and female study participants.

Data	Total Cohort		Males		Females	
	T1DM (*n* = 100)	Controls (*n* = 100)	T1DM (*n* = 44)	Controls (*n* = 44)	T1DM (*n* = 56)	Controls (*n* = 56)
Age (yr)	14.5 ± 2.7	14.3 ± 2.7	14.9 ± 2.8	14.6 ± 2.8	14.2 ± 2.5	14.1 ± 2.6
Father’s educational level						
Primary school	13 (15.3)	20 (20.6)	7 (19.4)	7 (16.7)	6 (12.2)	13 (23.6)
Middle school	7 (8.2)	18 (18.6)	2 (5.6)	9 (21.4)	5 (10.2)	9 (16.4)
High school	24 (28.2)	25 (25.8)	10 (27.8)	10 (23.8)	14 (28.6)	15 (27.3)
Diploma/Bachelor’s degree or higher	41 (48.2)	34 (35.0)	17 (47.2)	16 (38.1)	24 (49.0)	18 (32.7)
Mother’s educational level						
Primary school	21 (21.6)	17 (17.0)	7 (17.1)	6 (13.6)	14 (25.0)	11 (19.6)
Middle school	14 (14.4)	22 (22.0)	8 (19.5)	6 (13.6)	6 (10.7)	16 (28.6)
High school	19 (19.6)	21 (21.0)	9 (22.0)	12 (27.3)	10 (17.9)	9 (16.1)
Diploma/Bachelor’s degree or higher	43 (44.3)	40 (40.0)	17 (41.4)	20 (45.5)	26 (46.4)	20 (35.7)
Household income per month						
<300 USD	13 (13.0)	10 (10.3)	5 (11.4)	2 (4.8)	8 (14.3)	8 (14.5)
300 to <800 USD	33 (33.3)	49 (50.5)	15 (34.1)	23 (54.8)	18 (32.1)	26 (47.3)
800 to <1400 USD	23 (23.0)	19 (19.6)	11 (25.0)	8 (19.0)	12 (21.4)	11 (20.0)
≥1400 USD	31 (31.0)	19 (19.6)	13 (29.5)	9 (21.4)	18 (32.1)	10 (18.2)
Pubertal stage						
1	7 (7.0)	7 (7.0)	5 (11.4)	5 (11.4)	2 (3.6)	2 (3.6)
2	9 (9.0)	7 (7.0)	5 (11.4)	5 (11.4)	4 (7.1)	2 (3.6)
3	16 (16.0)	12 (12.0)	9 (20.5)	9 (20.5)	7 (12.5)	3 (5.4)
4	33 (33.0)	27 (27.0)	14 (31.8)	9 (20.5)	19 (33.9)	18 (32.1)
5	35 (35.0)	47 (47.0)	11 (25.0)	16 (36.4)	24 (42.9)	31 (55.4)
Height (cm)	156.3 ± 11.2	158.4 ± 11.7	161.3 ± 12.8	162.6 ± 13.9	152.4 ± 7.8	155.2 ± 8.5
Height Z-score	0.17 ± 1.20 **	0.67 ± 1.18	0.33 ± 1.30	0.72 ± 1.08	0.04 ± 1.11 *	0.63 ± 1.25
Weight (kg)	52.7 ± 14.6	54.1 ± 15.0	55.6 ± 15.5	56.4 ± 16.7	50.4 ± 13.3	52.4 ± 13.4
Weight Z-score	1.07 ± 1.65	1.35 ± 1.76	1.04 ± 1.71	1.30 ± 1.84	1.09 ±1.62	1.38 ± 1.70
BMI (kg/m^2^)	21.2 ± 4.0	21.2 ± 4.1	21.0 ± 3.9	20.9 ± 4.2	21.4 ± 4.2	21.5 ± 4.0
BMI Z-score	0.40 ± 1.08	0.44 ± 1.12	0.29 ± 1.18	0.35 ± 1.22	0.49 ± 1.00	0.52 ± 1.04
Lean mass (kg)	34.5 ± 9.2	35.0 ± 9.4	40.2 ± 10.1	40.6 ± 10.3	30.1 ± 5.3	30.6 ± 5.5
Total fat (%)	31.2 ± 9.9	31.6 ± 11.3	24.6 ± 8.3	24.3 ± 10.9	36.3 ± 7.7	37.3 ± 7.8
HbA1c (%)	8.89 ± 1.83 ***	5.24 ± 0.26	9.11 ± 2.10 ***	5.29 ± 0.26	8.71 ± 1.59 ***	5.20 ± 0.26
HbA1c (mmol/mol)	73.7 ± 20.0 ***	33.7 ± 2.9	76.1 ± 22.9 ***	34.3 ± 2.9	71.7 ± 17.3 ***	33.3 ± 2.8

Data were presented as *n* (%) and mean ± SD as appropriate; * *p* < 0.05, T1DM vs. control; ** *p* < 0.01, T1DM vs. controls; *** *p* < 0.001, T1DM vs. controls.

**Table 3 nutrients-14-05169-t003:** Macronutrient and micronutrient intake per day compared between those with and without type 1 diabetes for the entire study population, for male study participants, and for female study participants.

	All			Male			Female		
	T1D (*n* = 100)	Control (*n* = 100)	*p* Value	T1D (*n* = 44)	Control (*n* = 44)	*p* Value	T1D (*n* = 56)	Control (*n* = 56)	*p* Value
Energy intake (kcal/day)	1545 ± 410	1297 ± 440	<0.01	1662 ± 402	1468 ± 366	0.02	1454 ± 396	1163 ± 451	<0.01
Carbohydrate (%TE)	50.8 ± 6.8	46.2 ± 7.5	<0.01	52.3 ± 5.1	45.0 ± 8.3	<0.01	49.6 ± 7.7	47.2 ± 6.6	0.07
Fat (%TE)	32.4 ± 5.9	35.9 ± 6.4	<0.01	31.1 ± 4.5	37.2 ± 7.0	<0.01	33.4 ± 6.7	34.8 ± 5.8	0.23
Saturated fat (%TE)	10.2 ± 3.2	9.5 ± 3.2	0.21	10.5 ± 3.5	10.0 ± 3.4	0.44	10.0± 3.0	9.1 ± 3.1	0.14
Protein (%TE)	16.8 ± 2.6	17.9 ± 3.5	0.01	16.7 ± 2.3	17.8 ± 3.8	0.10	16.9 ± 2.9	18.0 ± 3.2	0.06
Cholesterol (mg/day)	292 (207–387)	299 (222–428)	0.53	293 (189–371)	327 (237–452)	0.15	289 (214–399)	285 (198–394)	0.64
Sugar (%TE) (natural and added sugar)	12.7 (9.0–17.9)	12.2 (8.0–17.9)	0.76	13.3 (8.1–18.7)	12.8 (9.2–16.9)	0.94	12.4 (9.3–17.8)	11.4 (6.7–19.0)	0.59
Calcium (mg/day)	474 (297–700)	329 (167–447)	<0.01	536 (363–781)	346 (215–454)	<0.01	462 (230–651)	280 (163–442)	<0.01
Sodium (mg/day)	2578 (2061–3235)	2687 (2000–3637)	0.74	2774 (1991–3310)	3111 (2230–4027)	0.08	2520 (2061–3177)	2322 (1891–3270)	0.19
Dietary fiber (g/1000 kcal)	3.2 (2.3–4.8)	3.0 (1.9–5.2)	0.67	2.9 (2.2–4.8)	2.4 (1.8–3.9)	0.06	3.5 (2.4–4.8)	3.7 (2.3–5.9)	0.32

Data were presented as mean ± SD and median (IQR) as appropriate; %TE: percentage of total energy.

**Table 4 nutrients-14-05169-t004:** The percentage of participants who did not meet nutritional recommendations compared between those with and without type 1 diabetes for the entire study population, for male study participants, and for female study participants.

	All			Male			Female		
	T1D (n = 100)	Control (n = 100)	*p* Value	T1D (n = 44)	Control (n = 44)	*p* Value	T1D (n = 56)	Control (n = 56)	*p* Value
Fat > 35%TE	29%	57%	<0.01	15.9%	70.5%	<0.01	39.3%	46.4%	0.45
Saturated fat > 10 %TE	50%	41%	0.20	52.3%	47.7%	0.67	48.2%	35.7%	0.18
Calcium < 800 mg/day	83%	94%	0.02	77.3%	88.6%	0.16	87.5%	98.2%	0.03
Sodium ≥ 2000 mg/day	81%	75%	0.31	75%	86.4%	0.18	85.7%	66.1%	0.02
Dietary fiber < 14 g/1000 kcal	97%	98%	0.65	97.7%	100%	0.32	96.4%	96.4%	1.00

Data were presented as percentage; %TE: percentage of total energy.

**Table 5 nutrients-14-05169-t005:** Physical activity during a typical week within the previous month compared between those with and without type 1 diabetes for the entire study population, for male study participants, and for female study participants.

	All			Male			Female		
	T1D (*n* = 100)	Control (*n* = 100)	*p* Value	T1D (*n* = 44)	Control (*n* = 44)	*p* Value	T1D (*n* = 56)	Control (*n* = 56)	*p* Value
Vigorous work (min/week)	0 (0–0)	0 (0–0)	0.39	0 (0–0)	0 (0–0)	0.79	0 (0–0)	0 (0–0)	0.23
Moderate work (min/week)	30 (0–90)	30 (0–70)	0.71	40 (0–105)	30 (0–70)	0.57	30 (0–88)	60 (13–101)	0.27
Travel (min/week)	0 (0–0)	0 (0–56)	0.01	0 (0–0)	0 (0–45)	0.04	0 (0–10)	0 (0–60)	0.10
Vigorous exercise (min/week)	30 (0–90)	45 (0–195)	0.21	60 (0–225)	150 (60–600)	<0.01	0 (0–60)	0 (0–83)	0.64
Moderate exercise (min/week)	30 (0–70)	20 (0–113)	0.95	30 (0–113)	0 (0–120)	0.80	30 (0–60)	25 (0–60)	0.82
Total MET value	940 (480–1975)	1120 (540–2815)	0.10	1520 (760–2835)	2460 (1050–6370)	<0.01	710 (315–1200)	860 (330–1390)	0.49
Overall physical activity (work, travel and exercise) (min/day)	25 (13–48)	34 (14–77)	0.04	40 (19–66)	65 (27–128)	0.01	21 (9–33)	28 (11–38)	0.22
Adequate physical activity ≥ 60 min/day	20 (20%)	30 (30%)	0.10	14 (31.8%)	23 (52.3%)	0.03	6 (10.7%)	7 (12.5%)	1.00
Moderate to vigorous exercise (min/week)	68 (33–180)	120 (23–360)	0.21	138 (60–323)	360 (71–720)	<0.01	60 (30–116)	50 (0–180)	0.72
Moderate to vigorous exercise ≥ 150 min/week	32 (32%)	48 (48%)	0.02	22 (50%)	29 (65.9%)	0.09	10 (17.9%)	19 (33.9%)	0.09

Data were presented as *n* (%) and median (IQR).

## Data Availability

The dataset used and/or analyzed during this study is available from the corresponding author upon reasonable request.

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
