# Peer review of "Dietary Intake and Physical Activity of Thai Children and Adolescents with Type 1 Diabetes Mellitus"

_nutrients, 2022, doi:10.3390/nu14235169_

Round 1

Reviewer 1 Report

This manuscript is generally clear, nicely structured, and well-written. The research has been conducted and analysed to a high standard, and the results are relevant and valuable for public health and patient care.  

I have just a few minor comments and suggestions.

Abstract, lines 15-16, 19, 21:  Please specify what the measures of variance are (SD?, 95% CI, Range?)

Introduction, Lines 40-42: Suggest to move Table 1 to the introduction, and refer to it here.

Line 54:  delete “normal”

Line 88: “…was postulated to be…” - - It’s not clear what this means (“is recommended to be…”? “is usually found to be…”?).

Results, general: The text can just summarise data that are in the tables. It is not necessary to report the same data (values) in the text, which just makes it more difficult to read.  

Tables 2 and 3, footnote: “as appropriate”.

Table 3: Currently only total sugar is reported. If possible, it would be useful to (also) separate out and report added or free sugars.

Table 4: Will be much easier to read if the table only gives the percent of subjects in the rows. There is no need to also report the (number of subjects).

Table 5: The “0” values are confusing. Are these data correct? It’s surprising to see a median of 0 but an IQR of 0-56 for example, and then these same median=0 values apparently also statistically significantly different. This may require further explanation. Is there perhaps a better way to analyse and present the data (for example log-transformed)?

Line 219 (also study limitations, Line 320): The T1DM subjects report consuming ~250 kcal/d more than controls, and similar or lower physical activity, but yet have the same mean BMI and body composition. This suggests the reported intake differences may not be valid reflection of habitual intakes. Authors should consider possible differences in reporting accuracy and completeness between the groups. There are sufficient individual anthropometric and physical activity data to easily calculate estimated individual energy requirements, and thus also estimate possible degree of under-reporting in each group.  It would not be surprising if the T1DM group more accurately record and report their intakes, given the attention on diet as part of their clinical management.

Lines 270-271: Suggest “…reduction in the rate of glucose absorption…”

Lines 272-274: I think these statements can be deleted, unless they specifically refer to populations with diabetes. The paragraph reads fine starting with the 3rd sentence, and the audience does not need reminding of the basis for dietary sodium guidelines.  

Line 325: A further limitation (perhaps) is that you were not able to separate out free or added sugars from total sugars.

Reviewer 2 Report

Article on the intake of selected nutrients (including intake compared to dietary recommendations) and physical activity in children and adolescents aged 5 to 20 years - comparison of the control group with the group with type 1 diabetes mellitus. The article was well prepared with extensive discussion of the results. Research and statistical methods were selected correctly. However, the article does not contain scientifically relevant content - although it is important, as the Authors write, and will be useful in developing strategies to improve treatment outcomes and the quality of life of Thai youth with T1D.

To consider:

-       - use words “dietary intake” or “physical activity” instead of “eating” or “exercise”,

-     - in the section " Body composition analysis " it is not described what was measured,

-     - in the section " Diabetes treatment, health insurance scheme, and socioeconomic status of participants " no mention of socio-demographic factors,

-      - whether chapters 2.4.-2.6. should not be before 2.2.-2.3. in the method description (in the order in which the results are presented).
